# Design and Characterization of an EEG-Hat for Reliable EEG Measurements

**DOI:** 10.3390/mi11070635

**Published:** 2020-06-28

**Authors:** Takumi Kawana, Yuri Yoshida, Yuta Kudo, Chiho Iwatani, Norihisa Miki

**Affiliations:** Department of Mechanical Engineering, Keio University, 3-14-1 Hiyoshi, Kohoku-ku, Yokohama, Kanagawa 223-8522, Japan; taku-23mp@keio.jp (T.K.); yuri.y0520@keio.jp (Y.Y.); yuta-tr_fivo@keio.jp (Y.K.); ciwatani@keio.jp (C.I.)

**Keywords:** electroencephalography, microneedle, dry electrode, headset

## Abstract

In this study, a new hat-type electroencephalogram (EEG) device with candle-like microneedle electrodes (CMEs), called an EEG-Hat, was designed and fabricated. CMEs are dry EEG electrodes that can measure high-quality EEG signals without skin treatment or conductive gels. One of the challenges in the measurement of high-quality EEG signals is the fixation of electrodes to the skin, i.e., the design of a good EEG headset. The CMEs were able to achieve good contact with the scalp for heads of different sizes and shapes, and the EEG-Hat has a shutter mechanism to separate the hair and ensure good contact between the CMEs and the scalp. Simultaneous measurement of EEG signals from five measurement points on the scalp was successfully conducted after a simple and brief setup process. The EEG-Hat is expected to contribute to the advancement of EEG research.

## 1. Introduction

Electroencephalography (EEG) is a method used for measuring electrical potential fluctuations originating from neural activity in the brain. EEG signals can be measured noninvasively from the surface of the head with a relatively simple system in comparison with other methods of measuring brain activity, such as functional magnetic resonance imaging (fMRI) and magnetoencephalography (MEG) [1,2]. Real-time measurement is a great advantage of EEG over near-infrared spectroscopy (NIRS), though the spatial resolution of EEG is inferior [3,4]. Although EEG has mainly been used for medical applications, it has recently become a major tool in the fields of psychology [5,6,7], cognitive science [8,9,10], and brain–computer interface (BCI) studies [11,12,13].

The number of EEG measurement sites and their positions depend on the application [14,15]. A 256-channel EEG system requires acquisition from the most positions possible [16]. The 10–20 system is another major standard used to determine the measurement sites [17]. Some applications require an even smaller number of measurement sites, which we consider is likely to become a mainstream requirement when EEG is used daily for BCI or the monitoring of the user’s mental or vital state. For such applications, measurement preparation must be minimally time-consuming and the whole EEG measurement system must be compact. 

In our previous work, we developed the candle-like microneedle electrode (CME) shown in Figure 1 [18]. The CME is composed of an array of protruding formations, which is a relatively thick pillar of 400 μm in diameter and 1000 μm in length. Each pillar has a sharp tip that is 200 μm in length and 200 μm in radius. The needle height is 200 μm, which does not reach the pain point or epidermis, and thus would not cause any infection. The sharp tips penetrate the upper surface of the skin, the high-impedance stratum corneum, eliminating the preparation that conventional wet electrodes require. The pillar at the base allows the electrodes to avoid hair, which enables successful EEG measurements from the scalp [19,20,21]. The CME is covered with parylene, which has sufficient mechanical strength and biocompatibility [22]. 

The fixation of electrodes is crucial for reducing the contact impedance between the electrode and the skin to obtain signals with a good signal-to-noise ratio [23]. In addition, it must provide the least stress possible to the user to minimize interference with the brain activities of interest. Table 1 summarizes the existing methods [24,25,26,27,28,29,30,31,32,33,34,35,36,37,38,39,40,41]. Net-type EEG devices are widely used in medical applications because they are capable of acquiring EEG signals from many positions with a good spatial resolution [27]. Cap-type devices are generally less bulky than net-type devices but can still measure a relatively large number of sites [31,40]. However, their appearance may not be appropriate for daily use. Several headgear-type devices have been commercialized, and these allow for the fixation of multiple electrodes [26,29,38,39]. They can be easily used and are sufficiently light to be comfortably wearable. As with other devices, avoiding hair may be a challenge. Some recent devices, such as the DSI-24 (Wearable Sensing, San Diego, CA, USA), have begun to overcome the challenges posed by pin-based dry electrodes [32]. However, their mechanical appearance should be improved for applications in daily life [42,43,44]. Headband-type devices are the simplest among the headsets; however, they only acquire signals from the forehead, where the measurement is not affected by the presence of hair but the noise level from the electrooculogram is large [45].

In the present study, we propose a hat-type EEG headset containing CMEs, called the EEG-Hat, which is simple to set up and can avoid interference due to hair. An overview of the design of the EEG-Hat is shown in Figure 2. The EEG-Hat consists of CME modules and a woven hat to integrate the modules. The CME modules allow the CMEs to be easily mounted and exchanged. As shown in Figure 3, the participant first puts on the hat and adjusts the distance between the empty modules and the scalp. Then, the CMEs are inserted into the modules and brought into contact with the scalp, where the hair is mechanically pushed away with a shutter mechanism. We presented the concept of the EEG-Hat first [46]. In this study, we refined the design to facilitate the setup and enhance the reliability in the measurement. Experiments and thorough analyses were conducted to further verify the effectiveness of the EEG-Hat.

## 2. Design and Fabrication

### 2.1. Module Design

The detailed design of the module is shown in Figure 4a. The module was designed to meet the following requirements:simple hair avoidanceadjustable contact force of the CMEsperpendicular contact between the CME pillars and the scalpsustained pressing of the CMEs.

Requirement 1 is the most challenging problem to overcome, as mentioned in Section 1. Requirement 2 affects the user comfort and the extent to which the CME microneedles penetrate the stratum corneum. Requirement 3 is necessary because the CME pillars properly penetrate the stratum corneum. For Requirement 3, the angle between the insertion part and the CME needs to be adjustable. Requirement 4 is also important to ensure that good contact is maintained between the electrodes and the scalp.

The overall function of the module is shown in Figure 4b. First, the insertion part is inserted into the connection part. Then, the shutter, which is housed in the connection part, is opened. During this process, the hair is pushed aside by the shutter, fulfilling Requirement 1. The shutter is pressed against the scalp by small screws, preventing the shutter from sliding over the hair. When the insertion part is pushed, the spring inside becomes compressed and presses the electrodes into the scalp. The insertion part can be fixed by slotting it into one of the grooves in the connection part, satisfying Requirement 4. There are seven grooves in the connection part. By changing which groove is used to fix the insertion part, the amount of compression in the spring can be adjusted, meaning the force with which the electrode is pressed into the scalp is adjustable, satisfying Requirement 2. Additionally, the electrode is mounted on the insertion part using a ball joint that can rotate freely according to the shape of the head, which satisfies Requirement 3. Thus, the electrodes can meet the scalp perpendicularly requirement without manual adjustment. The maximum rotation angle of the ball joint is 5°. The connection part is mounted onto an attachment part with screws to allow it to be fixed to the hat. Turning this screw adjusts the distance between the shutter and the scalp, enabling the hat to fit various head sizes. The grooves in the internal surface of the cylinder of the connection part are made by alternating the front and back of superimposed rings, as shown in Figure 4a. This mechanism makes it easier to create the grooves than to cut the inner wall surface of a cylinder.

The lead wire is connected to the electrode via the inside of the insertion part, which prevents the lead wire from hindering the movement of the shutter and contacting the electrode to the scalp. This was not the case for our prior work [46], where the design resulted in a low measurement yield and long setup time.

We highlighted two design parameters for the module, which are the shutter angle and the rotation angle of the ball joint. The shutter angle determines the separation of the hair and was experimentally found that 60° was the optimum [46]. The rotation angle, which is depicted in Figure 5, affects the contact between the electrode and the scalp. In this work, we investigated the rotation angles of 5° and 10° relative to the EEG signal quality and the experience of the participants. The larger rotation angles provide more opportunities to the CME to fit the shape of the head. However, when the rotation angle is large, the CME may approach the scalp with a tilted angle in the setup and provide pain to the participants.

### 2.2. Fabrication

Figure 2b,c shows the fabricated EEG-Hat and one of its modules. Most of the parts of the module were fabricated by cutting aluminum and acrylic plates with a wire-electrical discharge machine (MV1200S, Mitsubishi Electric Corp., Tokyo, Japan) and a numerical control (NC) milling machine (KE55, Makino Inc., Tokyo, Japan). The springs and acrylic balls for the ball joint were ready-made products. The parameters for the spring were selected experimentally, as described below. The diameter of the ball was 10 mm. The edges of the shutters were covered with plastic tapes, as shown in Figure 2c, and does not hurt the scalp. The hat was designed and provided by a hatmaker (Double Ribbon Co. Ltd., Tokyo, Japan). There were five holes in the hat for the attachment of the modules, and the holes were located at approximately F4, F3, Cz, O2, and O1 in the international 10–20 system.

## 3. Experimental Procedures

### 3.1. Quantification of Contact Force and Hair Separation

During the acquisition of EEG signals, it is necessary to separate the hair and expose the scalp such that the electrodes can be put in contact with the scalp. The EEG-Hat with CMEs was designed to perform this hair avoidance task mechanically. Therefore, it is important to clarify how much hair separation and contact force was necessary for the EEG measurement with CMEs. This was evaluated by measuring the scalp–electrode impedance relative to the amount of hair. The amount of hair was quantified as the ratio of the area occupied by the hair to the area of the electrode, and this parameter was termed the “hair occupancy rate,” defined as:
(1)$$O_{hair}=\frac{A_{hair}}{A_{total}},$$
where $A_{hair}$ is the area occupied by the hair, and $A_{total}$ is the area where the electrode is in contact with the scalp. $A_{hair}$ and $A_{total}$ were obtained by analyzing photographs of the measurement location with image editing software (Adobe Photoshop CC 2018, 19.1.9, Adobe, San Jose, CA, USA). A grayscale conversion was applied to the photographs, and $A_{hair}$ and $A_{total}$ were obtained in units of pixels, as shown in Figure 6.

In this experiment, the scalp–electrode impedance was used as the evaluation index. However, this parameter is highly subject-dependent, in part because of the variability in human dermal characteristics [47,48,49]. For example, harder skin requires more force to lower the impedance. Therefore, to eliminate the influence of the variable dermal characteristics and contact force between different subjects, the baseline impedance was normalized, as described below. First, the hair was separated as much as possible. Then, the CME was pressed against the scalp until the impedance reached 30 kΩ, at which point the contact force was recorded. With this method, the relationship between the hair occupancy rate and the impedance was determined, along with the contact force necessary to perform the EEG measurements with CMEs. The contact force was obtained manually using a digital force gauge (ZP-50N, IMADA, Aichi, Japan), and the integer part of the displayed force was recorded. After that, the target force was kept fixed, and only the amount of hair was changed to determine its effect on the impedance. To obtain the hair occupancy rate, every time the hair was adjusted, a photograph was taken of the measurement point. The amount of hair was changed 11 times, including during the first force recording, and each time the impedance and a photograph were obtained. The impedance was recorded using an impedance meter (SIGGI II, Easycap, BRAIN VISION UK Ltd., London, UK). The subjects were four men and a woman in their twenties.

### 3.2. Determination of the Shutter Angle

This experiment was conducted to assess whether the shutter mechanism could separate hair enough for the EEG measurements. It was assumed that the degree to which the shutter could separate hair was determined by the shutter angle, as defined in Figure 7. Three shutters were fabricated with shutter angles *θ* of 50°, 55°, and 60°. These three shutter angles were selected by considering the size of the module and the ease of opening the shutter. The long shutter was unsuitable for a compact appearance of the device. Figure 8a shows the relationship between the parameters of the shutter. The shutter height *y* in Figure 8a was found using the following Equation (2) with shutter angle *θ*:(2)$$y=12\tan\theta +\frac{1.5}{\cos\theta }$$

According to Equation (2), when the shutter angle *θ* is over 65°, the shutter height *y* becomes over 30 mm and starts to increase rapidly. Therefore, we decided that the shutter angle *θ* should be below 65°. Moreover, the shutter opens using the force applied by the contact with the insertion part, as shown in Figure 8b. In Figure 8b, when the shutter angle *θ* was below 45°, the horizontal component of *f* became larger than the vertical component, and opening the shutter became difficult. Therefore, shutter angles *θ* over 45° were selected. We tested the shutter angles *θ* of 50°, 55°, and 60° to find the most suitable one.

For each shutter angle, the hair occupancy rate was measured each time the shutter was opened, with hair angles *ϕ* of 90°, 45°, and 0°. The hair angle *ϕ* is the angle between the direction in which the shutter opens and the direction of the hair, as shown in Figure 7. It was estimated that the shutter was most likely to separate hair when *ϕ* was 90°. Since the actual hair direction of the subjects might vary, three different hair angles were investigated in this experiment. This experiment was conducted with an artificial wig for medical use (Propia, Tokyo, Japan) instead of human hair.

### 3.3. Rotation Angle of the Ball Joint

The participants were eight healthy men and two healthy women in their twenties. EEG measurements were conducted while the participant’s eyes were closed, and the signals were acquired at O1 (left occipital part) in the international 10–20 system. For all the participants, two rotation angles of 5° and 10° were applied with an EEG recording time of 30 s, where 5° and then 10° was tested for the first half of the participants and the order was reversed for the other half. To make the test conditions identical except for the rotation angle, we separated the hair manually under visual observation. The pressing force of the electrode was set to be identical for 5° and 10°. After the measurements using the two rotation angles, the participants were requested to answer which angle was less painful or bothersome.

### 3.4. EEG Measurement with the EEG-Hat

The EEG measurement performance of the EEG-Hat was then evaluated. EEG measurements were conducted while the participant’s eyes were open and while they were closed, and signals were acquired at F4, F3, Cz, O1, and O2 in the international 10–20 system. The reference and ground were at A1 and A2. The five CMEs mounted in the EEG-Hat were connected to a polygraph system (RMT-1000, Nihon Kohden, Tokyo, Japan). Labchart 7 (ADInstruments, Sydney, Australia) was used as the EEG recording software. The acquired data were filtered using a notch filter at 50 Hz and a bandpass filter with a pass band of 0.5–30 Hz. The subjects were four healthy men in their twenties. During the EEG measurements, the subjects had their eyes open during the first 30 s, and then closed their eyes for 5 min. The scalp–electrode impedance was measured at the start and the end of the measurement using the impedance meter (SIGGI II, Easycap, BRAIN VISION UK Ltd., London, UK). The experiment was conducted with the subjects seated and they maintained a steady position as much as possible during the experiment. The environment of the measurement was non-shielded. The hair was mechanically separated using the shutters in the modules. The setup time of each channel was also measured, which included the time for the CME to be mounted on the insertion part, for the insertion part to be inserted into the connection part (see Step 2 in Figure 3), and for the impedance to reach a stable level below 200 kΩ. When the impedance did not decrease below 200 kΩ, the insertion part was removed from the connection part and then inserted into the connection part again. This iteration is included in the setup time.

## 4. Results

### 4.1. Quantification of the Contact Force and Hair Separation

Figure 9 shows the scalp–electrode impedance plotted against the hair occupancy rate for the five subjects, along with the subject-dependent contact force obtained in the normalization process described in Section 3.1. The recorded contact force for each subject is in the graph. The contact forces were in the range of 8–15 N. The impedance meter we used was not capable of measuring more than 200 kΩ. Impedances exceeding 200 kΩ are plotted as “over 200” in Figure 9.

### 4.2. Determination of the Shutter Angle

Figure 10 shows the results of the shutter angle test. The hair occupancy rate is plotted against the shutter angle for different hair angles. The hair occupancy rate was considered to be the average of five measurements for each combination of shutter angle and hair angle. At *ϕ* = 90°, all shutter angles reduced the impedance to below 50%. At *ϕ* = 45°, the hair occupancy ratio at a shutter angle of 50° was approximately 10% higher than that at shutter angles of 55° and 60°. At *ϕ* = 0°, all shutter angles yielded hair occupancy rates of over 80%.

### 4.3. The Rotation Angle of the Ball Joint

We adopted the signal-to-noise ratio (*SNR*) to represent the signal quality, where *SNR* was calculated using Equation (3):(3)$$SNR=10\log_{10}\frac{\bar{PSD_{\alpha }}}{\bar{PSD_{s-\alpha }}},$$
where *PSD* is the power spectral density of the raw EEG data. In this experiment, the original EEG data was divided into epochs with 2048 data points with a 50% overlap. The *PSD* estimate was applied to each epoch and all epochs were averaged. In Equation (3), $\bar{PSD_{\alpha }}$ is the *PSD* for the alpha band, and $\bar{PSD_{s-\alpha }}$ is the *PSD* for all frequency bands other than the alpha band. In this experiment, the *PSD* was obtained by Labchart 7 (ADInstruments, Sydney, Australia).

Table 2 shows the results. There was no significant difference in the average *SNR* between the rotation angles of 5° and 10°. The measurement order did not make a significant difference either.

No participants reported severe pain. Six participants answered that the rotation angle of 10° provided more pain.

### 4.4. EEG Measurement with the EEG-Hat

The recorded EEG signals were analyzed using MATLAB R2020a (MathWorks, Natick, MA, USA) to calculate their *PSD*s and *SNR*s. In this experiment, to eliminate biological artefacts, such as heartbeat and breathing, only the frequency range of 4–30 Hz in the recorded EEG signals was considered [50,51].

Table 3 gives the *SNR*s obtained while the participant’s eyes were open and while they were closed. The *SNR* was calculated to verify that the EEG-Hat recorded meaningful EEG signals. EEG data spanning 30 s from 5 s after the start of the measurement were analyzed for both while the participant’s eyes were open and while they were closed. A comparison of the *SNR*s obtained from each channel for each subject demonstrates that a higher *SNR* was obtained while the participant’s eyes were closed than while they were open for all signals, except that from channel O2 for Subject 4.

Figure 11 shows the *PSD* for each channel for 5 min during eye closure. From this 5 min *PSD*, the stability of the EEG measurement with the EEG-Hat was obtained. The PSD for each channel for each subject was normalized relative to its integral over the plotted range and averaged over all subjects. Large peaks corresponding to the alpha band (8–13 Hz) were observed in channels F3, Cz, O2, and O1. In F4, although the peak corresponding to the alpha band was not clear, the power in the alpha band was higher than that for frequencies exceeding 13 Hz.

Figure 12 shows the scalp–electrode impedance for each channel at the beginning and the end of the experiments. The measurement limit of the impedance meter was 200 kΩ. Since we could acquire sufficiently good EEG measurements when the impedance was below 200 kΩ, we set the acceptable impedance to be 200 kΩ in the experiments. The × mark means that the impedance was over 200 kΩ. The increase of impedance was less 50 kΩ for 13 out of 20 channels and it was over 50 kΩ for 7 out of 20 channels. For Cz, an increase of over 50 kΩ was observed for all the participants. The impedance at O2 for all the participants remained below 100 kΩ.

The average setup time was 179 ± 131 s (*n* = 20). Compared to our prior work, which took 20 min for each channel, the setup time was successfully reduced. Seven out of 20 channels were set up successfully, i.e., the impedance reached the plateau below 200 kΩ, at the first attempt. In these cases, the setup time was 48 ± 20 s (*n* = 7). For the other 13 channels, the impedance did not go down below 200 kΩ at the first attempt and the setup was iterated until the setup was successful. For these cases, the setup time was 250 ± 108 s (*n* = 13).

Visual observations did not show any damage to the scalp. No participants reported severe pain during the 5.5 min EEG measurement.

## 5. Discussion

### 5.1. Quantification of the Contact Force and Hair Separation

For non-critical EEG measurements with dry-electrode EEG systems, an impedance of approximately 100 kΩ is considered to provide sufficient measurement performance [52]. Therefore, in this experiment, impedances of 100 kΩ were deemed appropriate for stable EEG measurements. From Figure 9, impedances of over 100 kΩ were obtained from hair occupancy ratios of 80% or more. The percentage of impedance measurements of 100 kΩ or less was 97% for hair occupancy ratios below 80% (*n* = 30) and 32% for those above 80% (*n* = 25). These results indicate that for the CMEs to achieve an impedance that was sufficient for good measurement performance, it was necessary to separate the hair until the hair occupancy ratio was below 80%. Additionally, it was found that a pressing force in the range of 8–15 N was sufficient for EEG measurements with CMEs. From the result of the pressing force, the spring in the insertion part was designed to have a spring constant of 1.36 N/mm. The designed modules can compress the spring in increments of 1.5 mm up to 12 mm. With this design, the EEG-Hat could provide a contact force ranging from 0 to 16.3 N. In this experiment, participants 1 and 4 reported discomfort as the experiment progressed. The forces in the above range were recorded when the hair was manually separated as much as possible and the impedance was below 30 kΩ. Given that an impedance of approximately 100 kΩ is sufficient for EEG measurements with dry-electrodes, including CMEs [52], for the EEG measurements with the EEG-Hat, it was not necessary to press the electrodes until the participants felt discomfort when the hair was properly separated by the shutter. For the EEG-Hat, the subjects could select a force for which they did not feel discomfort within the range of 0 to 16.3 N by changing the shrinkage of the spring in the insertion part.

### 5.2. Determination of the Shutter Angle

Based on the results discussed in Section 5.1, the maximum acceptable hair occupancy ratio to enable the CMEs to yield stable EEG measurements was determined to be 80%. Thus, at hair angles of 90° and 45°, all shutter angles made it possible to stably acquire EEG signals. Moreover, the measured hair occupancies at *ϕ* = 45° indicated that EEG measurements with a shutter angle of 50° or less would tend to be less stable than at higher shutter angles. At *ϕ* = 0°, the EEG measurements would be unstable regardless of the shutter angle. Therefore, the hair angle must not be 0°, and this can be solved by rotating the connection part to adjust the angle as needed. From these results, the shutter angle does not significantly influence the hair occupancy ratio in the considered range of *θ* = 50° to 60°. However, from the results obtained at *ϕ* = 45°, *θ* = 50° appeared to yield the least reliable results. Furthermore, a larger shutter angle was achieved with a smaller device and allowed for the shutter to be opened more easily. For all of these reasons, *θ* = 60° was selected for the design of the developed EEG-Hat.

### 5.3. The Rotation Angle of the Ball Joint

As we described in Section 4.3, the rotation angles of 5° and 10° did not result in a change of the signal quality. No subjects answered that 5° was more painful than 10°. Subject 2 reported a slight movement of the electrode at the rotation angle of 10°. We suspected that this was because the degree of freedom of the ball joint increased more than necessary. Subject 3 said that he felt a thornier feeling at the rotation angle of 10° than 5°. The thorny feeling is caused when the corner of the electrode holder contacted the scalp. From the above discussions, we concluded that the rotation angle of 5° was sufficiently good for the EEG-Hat.

### 5.4. EEG Measurements Using the EEG-Hat

The current design of the EEG-Hat was experimentally verified to require a short setup time. It has been known since the 1930s that the activity in the alpha band is higher while the participant’s eyes were closed than while they were open [53]. This appears particularly prominently in the posterior region [53]. Thus, the increase in the *SNR* while participant’s eyes were open and then closed obtained in the present study demonstrates that the EEG-Hat was able to detect alpha waves, verifying the successful acquisition of EEG signals.

The results reported in Table 3 indicate that the EEG signals were successfully recorded for all channels for Subjects 1–3 and for F4, F3, Cz, and O1 for Subject 4. The increase in the *SNR* for O2 and O1 for Subjects 1–3 was larger than that for the other channels, which is consistent with past reports [54]. For Subject 4, the *SNR* for O1 increased, whereas that for O2 did not. This indicates that only channel O2 for subject 4 had problems, e.g., a low pressing force, unstable contact between the CMEs and the scalp, and insufficient hair avoidance. Except for this one channel, the EEG-Hat was able to successfully acquire EEG signals.

The results shown in Figure 11 indicate that these channels recorded signals in the alpha band (8–13 Hz) accurately and continuously for 5 min. Thus, the EEG measurements were stable for these five channels. The power in the alpha band for F4 was lower than that for the other channels. From this, it was inferred that F4 had more artefacts than the other channels. Because this was a trend seen only for F4, it might have been caused by artefacts appearing only in a single channel, such as muscle or electrode-pop artefacts [55,56]. The EEG-Hat functions in such a way that independent modules apply a force to each electrode. This mechanism enables a simple setup procedure and mechanical hair avoidance. However, this mechanism was determined to cause nonuniform deformations of the hat’s surface, and the deformations were concentrated around F4. Thus, in future versions of the EEG-Hat, it will be necessary to ensure a more even contact force for all the electrodes.

According to Figure 12, Cz was the only channel in which the impedance increased by 50 kΩ or more for all the participants. Each channel receives a reaction force against the pressing force of the electrode to the scalp, which resulted in the deformation of the hat. The deformation at the position of Cz was considered to be the largest among all the measurement positions. In other positions, during the 5 min EEG measurement, the increase of the impedance was acceptable.

## 6. Conclusions

A new hat-type EEG device, which is termed the EEG-Hat, was developed in this study. The EEG-Hat was capable of separating hairs and securing contact between the candle-like microneedle electrode (CME) and the scalp. The amount of hair was quantified as the hair occupancy rate, which was computed through image analysis, and it was demonstrated that a hair occupancy ratio of 80% or less and a pressing force of 8–15 N were necessary to ensure sufficient performance of the CMEs during EEG measurements. A shutter mechanism was designed to reduce the hair occupancy and equipped onto the EEG-Hat and was tested to determine the appropriate parameters to ensure sufficient EEG measurement accuracy. The rotation angles of the ball joint to adjust the angles of the CME to the scalp were tested and the rotation angle of 5° was experimentally selected as the suitable parameter after taking the signal-to-noise ratio and user experiences into consideration. EEG signals were stably acquired in five channels without manual hair separation, while the setup time was drastically reduced relative to our prior work. The fixation mechanism for EEG electrodes, such as an EEG-Hat, will play a critical role in the advancement of EEG research. The characterization of such mechanisms relative to the *SNR,* the contact impedance, the setup time, and the user experience can be a good design guideline. These proposed guidelines will contribute to the development of comfortable and reliable EEG devices.

## Figures and Tables

**Figure 1 micromachines-11-00635-f001:**
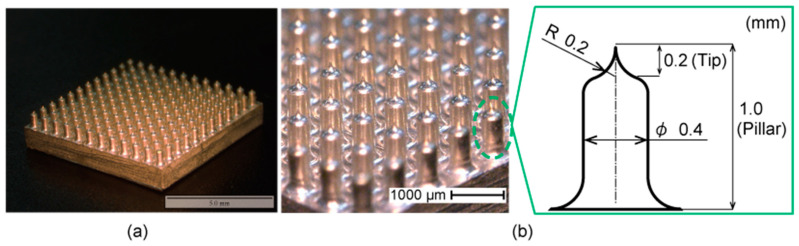
(**a**) Overview appearance of a candle-like microneedle electrode (CME). (**b**) An enlarged view of the array and the parameters of a pillar and its sharp tip. The wide bases allow them to avoid hair and the sharp tips penetrate the stratum corneum to reduce the contact impedance.

**Figure 2 micromachines-11-00635-f002:**
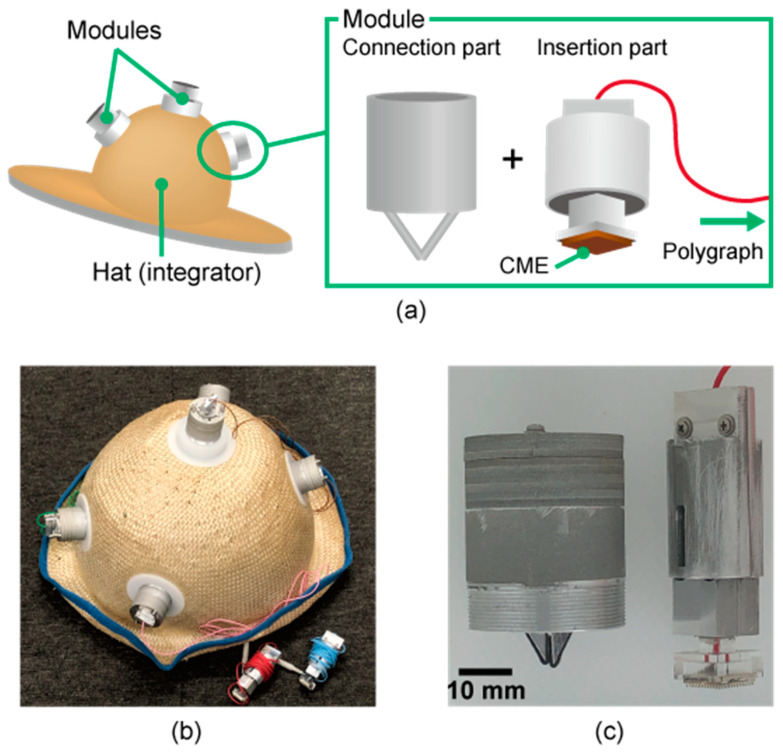
(**a**) Overview of the EEG-Hat. The EEG-Hat consists of independent modules and a woven hat that integrates the modules. Each module consists of a connection part and an insertion part. (**b**) The overall appearance of the EEG-Hat. Five modules are connected to the hat with white attachment parts. (**c**) Fabricated module. The connection part has a screw on the bottom to connect it to the attachment part. The CMEs are mounted in the insertion part.

**Figure 3 micromachines-11-00635-f003:**
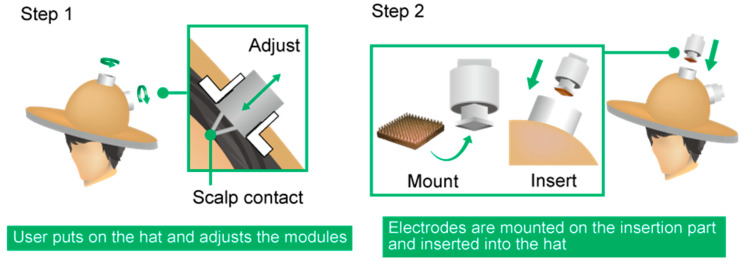
Use of the EEG-Hat. The preparation of the EEG-Hat for measurement involves two steps. The module can be used to adjust the hat to fit various head sizes and shapes.

**Figure 4 micromachines-11-00635-f004:**
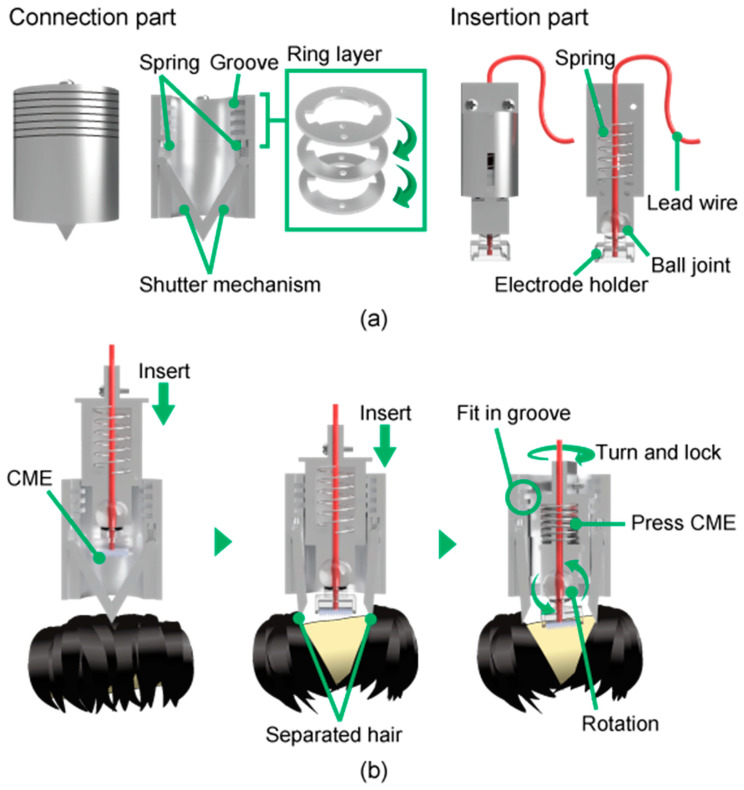
(**a**) Detailed design of the module. The shutter mechanism is housed in the connection part. CMEs are mounted in the insertion part and connected to a polygraph via the lead wire. (**b**) The overall function of the module. Hair is mechanically separated by the shutter mechanism. The contact between the CME and the scalp is maintained by the springs and the insertion part slotting into one of the grooves in the connection part.

**Figure 5 micromachines-11-00635-f005:**
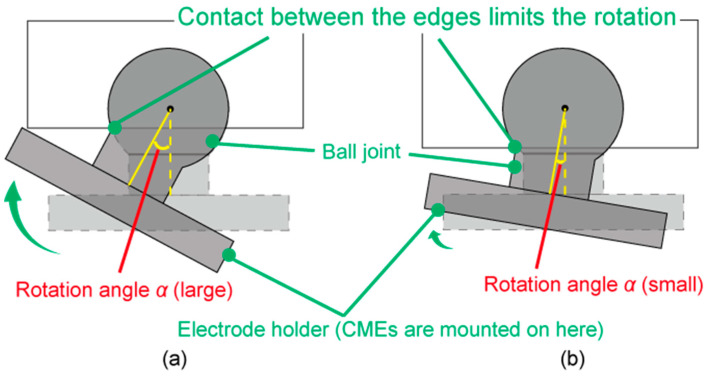
Images of the ball joint with (**a**) a large rotation angle α and (**b**) a small rotation angle α. The rotation stops when the edges of the ball joint hit the holder.

**Figure 6 micromachines-11-00635-f006:**
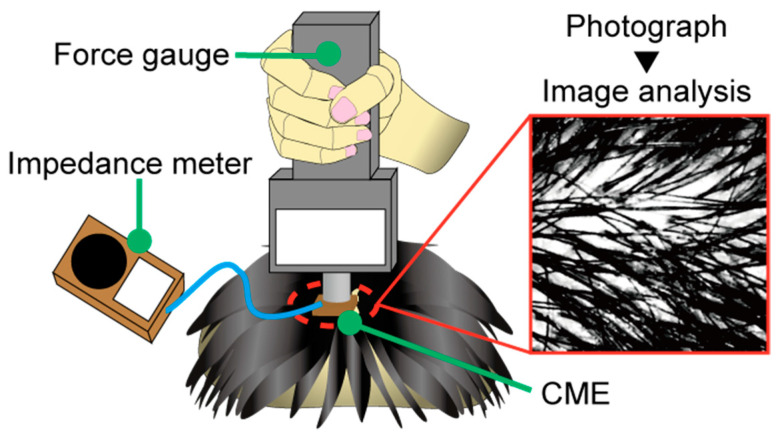
Image of the hair occupancy test. The CME was manually pressed against the scalp with a force gauge. The impedance was recorded with a meter. An example of an analyzed image of the hair occupancy rate is shown.

**Figure 7 micromachines-11-00635-f007:**
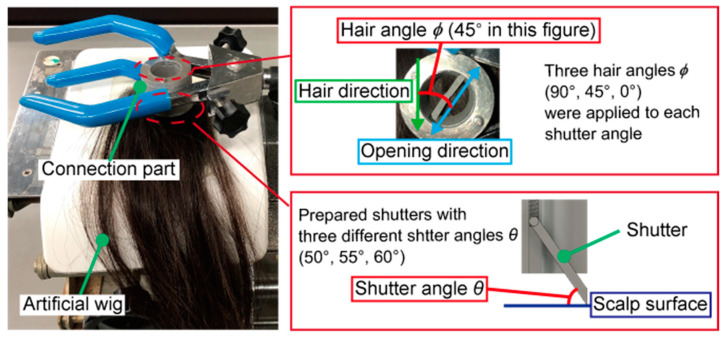
Photograph of the shutter angle test and definitions of the angles *θ* and *ϕ*. The amount of hair near the shutter was randomly changed by hand each time.

**Figure 8 micromachines-11-00635-f008:**
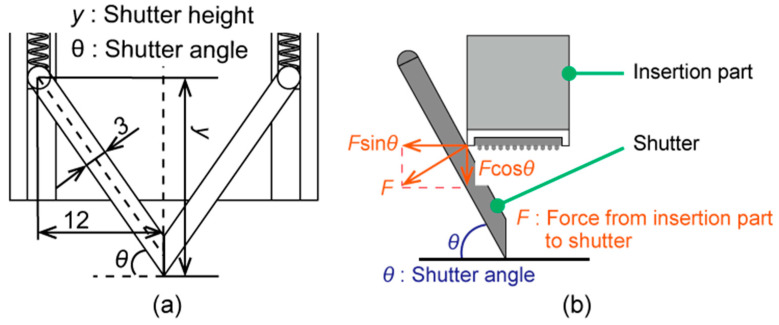
(**a**) The relationship among the parameters of the shutter. Shutter height *y* is obtained using geometry in terms of the shutter angle *θ.* (**b**) The simple model of the relationship between the forces that are related to opening the shutter.

**Figure 9 micromachines-11-00635-f009:**
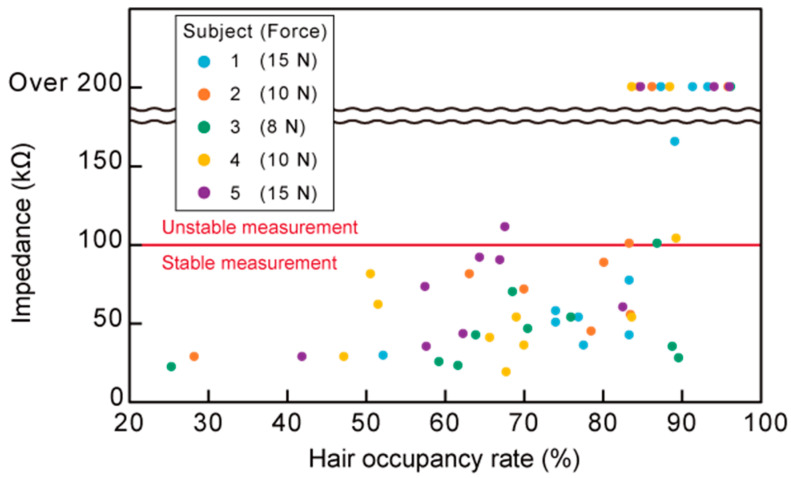
Measured impedance for various hair occupancy rates and contact forces for five different subjects. An impedance of 100 kΩ or less was considered sufficient to realize a stable EEG measurement.

**Figure 10 micromachines-11-00635-f010:**
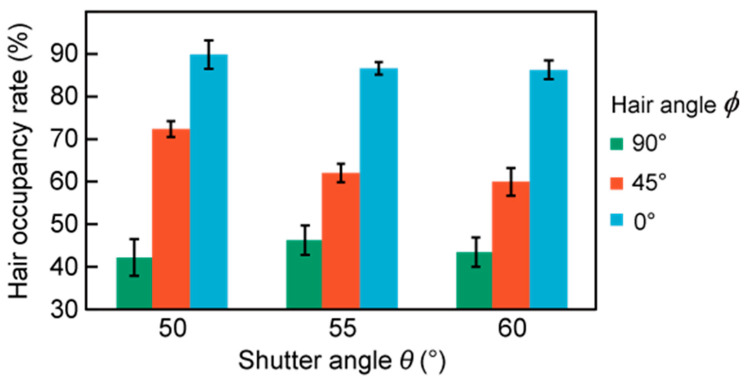
Hair occupancy ratio at different shutter angles and hair angles. The shutter angle and hair angle are defined in Figure 8.

**Figure 11 micromachines-11-00635-f011:**
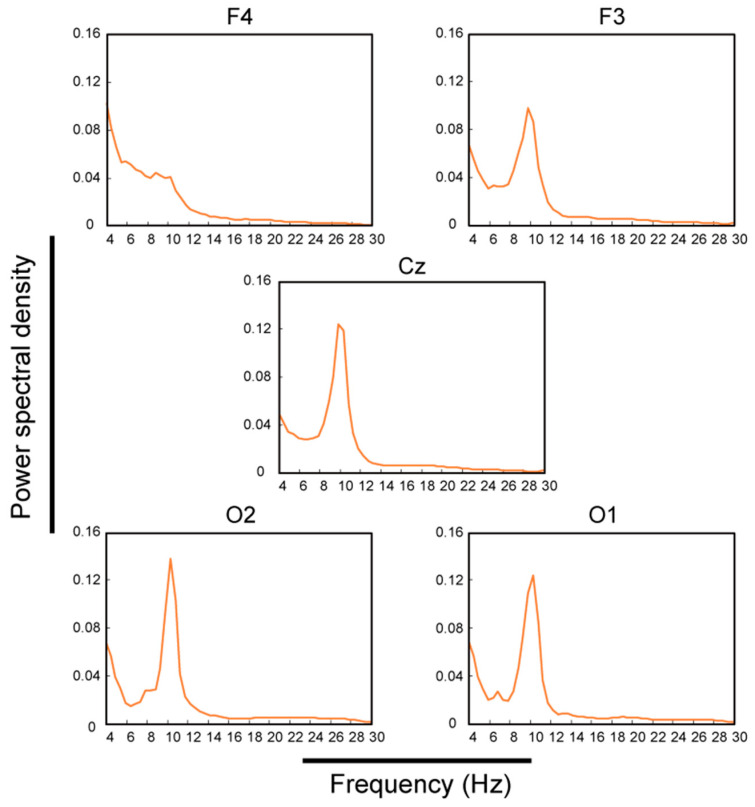
Power spectral density (*PSD*) for each channel averaged over all subjects for channels F3, F4, Cz, O2, and O1. The alpha band spanned from 8 to 13 Hz.

**Figure 12 micromachines-11-00635-f012:**
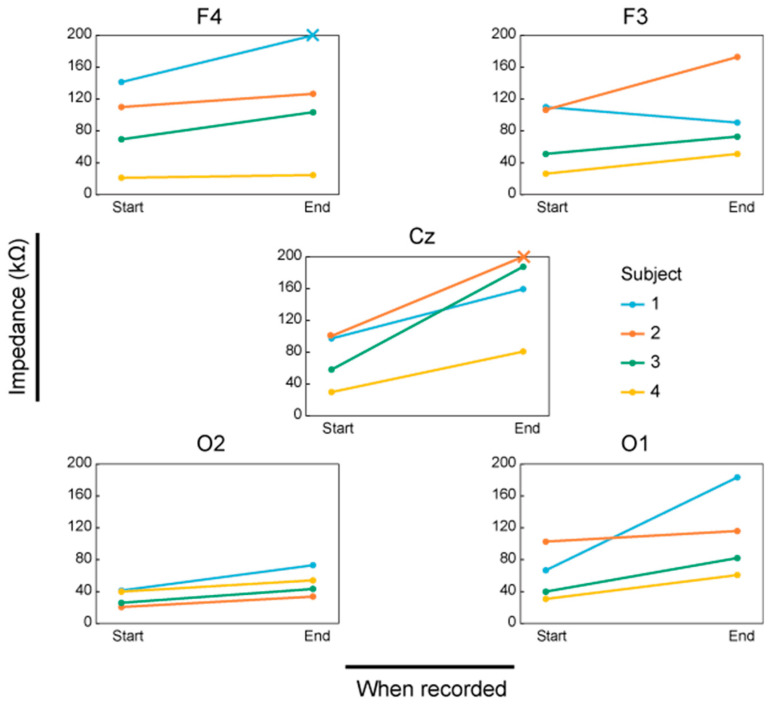
The scalp–electrode impedance of each channel at the beginning and the end of the measurement. An × mark shows an impedance of above 200 kΩ.

**Table 1 micromachines-11-00635-t001:** Existing methods used in current electroencephalogram (EEG) devices.

Appearance	Products (Company) [Reference]	Number of Channels	Electrode	Example of Application [Reference]
Net-type	GES400 (Electrical Geodesics) [47]	256	Saline	Clinical [27] High-density EEG [35]
	R-Net (Brain Products) (Press release) [31]	128	Saline	
Cap-type	actiCAP (Brain Products) [31]	256	Dry	Virtual reality [40] Neurofeedback [28] Brain imaging [30]
	g.Nautilus research (g.tec) [29]	64	Saline or Gel	
	Quick-Cap (Compumedics Neuroscan) [38]	32	Gel	
Headgear-type	Ultracortex Mark Ⅳ (Open BCI) [38]	32	Dry	BCI [41] Virtual reality [25] Game [26] Mental health [36] Driver’s state [33]
	32 Trilobite (Mindo) [29]	32	Dry	
	Quick-30 (CGX) [39]	30	Dry	
	DSI-24 (Wearable Sensing) [32]	24	Dry	
	EPOC+ (Emotiv) [29]	14	Saline	
	BR8+ (BRI) [29]	8	Dry	
Headband-type	Muse (InteraXon) [34]	2	Dry	Meditation [24]
	MindWave Mobile2 (NeuroSky) [34]	1	Dry	Education [37]

**Table 2 micromachines-11-00635-t002:** *SNR* and experiences of the participants with respect to the rotation angle.

Participant	*SNR* of 5°	*SNR* of 10°	Measurement Order	Which Was More Painful?
1	2.5	−0.4	5° → 10°	10°
2	9.4	−5.9	5° → 10°	10°
3	7.9	3.5	5° → 10°	10°
4	9.2	9.0	5° → 10°	Did not feel any difference
5	10	6.8	5° → 10°	Slightly 10°
6	14	13	10° → 5°	Did not feel any difference
7	−0.2	−1.3	10° → 5°	10°
8	6.2	7.9	10° → 5°	Did not feel any difference
9	12	−1.2	10° → 5°	Did not feel any difference
10	3.6	1.8	10° → 5°	Slightly 10°, almost no difference
mean ± SD	7.5 ± 4.5	3.4 ± 5.6		

**Table 3 micromachines-11-00635-t003:** *SNR* for while the participant’s eyes were open and while they were closed.

Participant		F4	F3	Cz	O2	O1
1	Open	−5.5	−4.6	−5.5	0.4	−3.6
	Close	−1.1	0.2	−3.8	5.0	2.1
2	Open	−6.5	−5.1	0.1	−8.9	−4.2
	Close	−5.7	3.2	3.8	6.0	5.3
3	Open	−3.6	−6.3	−0.9	−3.6	−3.0
	Close	1.2	−5.3	1.5	3.0	5.8
4	Open	−5.9	−2.5	−3.4	−6.6	−4.6
	Close	−1.5	4.6	2.8	−6.6	3.4

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
