# Peer review of "Design and Characterization of an EEG-Hat for Reliable EEG Measurements"

_micromachines, 2020, doi:10.3390/mi11070635_

Round 1

Reviewer 1 Report

The manuscript "EEG-Hat with Candle-like Microneedle Electrodes", by Takumi Kawana et al. reports the design and the fabrication of a new hat-type electroencephalogram (EEG) device with candle-like microneedle electrodes (CMEs).
It is an interesting, well-written paper.

A major concern is that a similar report has already been published under the same title.
Kawana, T., Yoshida, Y., Kudo, Y., & Miki, N. (2019). "EEG-Hat with Candle-like Microneedle Electrode". 2019, 41st Annual International Conference of the IEEE Engineering in Medicine and Biology Society (EMBC). doi:10.1109/embc.2019.8857477.

Although the current work presents an extended and enreached results. The EEG-Hat itself, does not seem to have any improvement in comparison to the paper mentioned above. 

Therefore, and for the present paper the authors should, 

  1. clarify possible improvements/innovations in the EEG-Hat
  2. if the EEG-Hat is the same technology, then please explain how do the new results justify the originality of the paper
  3. consider to adjust the title
  4. cite the above mention paper 

Due to our concers above, we have chosen "no answer" for both the originality and the overall merit of the paper. 

In addition, since no biocompatibility study is presented and since the eventual application of these electrodes are for humans it could be interesting for the readers if the Authors can provide information of biocompatibility the materials used. 

Author Response

We greatly appreciate your comment. We presented the concept of the EEG-Hat in the international conference (EMBC 2019). Since then, we have refined the design of the EEG-Hat and conducted experiments to verify the effectiveness. We added the following sentences in 1. INTRODUCTION

1. Introduction
Line 67:

We firstly presented the concept of the EEG-Hat [46]. In this paper, we refined the design to facilitate the setup and enhance the reliability in the measurement. Experiments and thorough analyses were conducted to further verified the effectiveness of the EEG-Hat.

Therefore, and for the present paper the authors should, 

  1. clarify possible improvements/innovations in the EEG-Hat

The major improvements of the EEG-Hat relate to:

1) The insertion part with the lead wire

2) The rotation angle of the ball joint.

1) The insertion part with the lead wire

Figure R1 (a) shows the electrode holder in the previous report. The lead wire that was connected to the electrode hindered the movement of the shutter and the electrode. In the current design (see Figure R1 (b)), the lead wire is connected via the inside of the insertion part. This design change culminated in high-yielding measurement and short setup time. We measured the setup time to highlight the effectiveness of the current design.
We added explanation in the manuscript as follows:

2.1. Module design
Line 105:

The lead wire is connected to the electrode via the inside of the insertion part, which prevented the lead wire from hindering movement of the shutter and contact of the electrode to the scalp. It was not the case for our prior work [46], where the design resulted in low measurement yield and long setup time.

3.4. EEG measurement with the EEG-Hat (in 3. Experimental procedures)
Line 201:

 The setup time of each channel was also measured, which included the time for the CME to be mounted on the insertion part, for the insertion part to be inserted into the connection part (see step 2 in Figure 3), and for the impedance to reach a stable level below 200 kΩ. When the impedance did not decrease below 200 kΩ, the insertion part was removed from connection part and then inserted into the connection part again. This iteration is included in the setup time.

4.4. EEG measurement with the EEG-Hat (in 4. Results)
Line 255:

Average setup time was 179 131 s (N =20). Compared to our prior work, which took 20 min for each channel, the setup time was successfully reduced. 7 out of 20 channels were successfully setup, i.e., the impedance reached the plateau below 200 kΩ, at the first attempt. In these cases, the setup time was 48 20 s. In the cases of the other 13 channels, the impedance did not go down below 200 kΩ at the first attempt and the setup was iterated until the setup went successful. For these cases, the setup time was 250 108 seconds (N=13).

5.4. EEG measurement with the EEG-Hat (in 5. Discussion)

Line 306:

 The current design of the EEG-Hat was experimentally verified to require short setup time.

2)The rotation angle of the ball joint

In the previous report, we investigated the shutter angle. The shutter angle determines the separation of the hairs. The shutter angle of 60°was experimentally found to be suitable. We added discussion on the shutter angle (see Section 3.2). In addition to this, in this paper, we looked into the rotation angle of the ball joint. The rotation angle affects the contact between the electrode and the scalp.

We add the following sentences:

2.1. Module design
Line 109:

We highlighted two design parameters for the module, which are the shutter angle and the rotation angle of the ball joint. The shutter angle determines the separation of the hairs and was experimentally found that 60° was the optimum [46]. The rotation angle, which is depicted in Figure 5, affects the contact between the electrode and the scalp. In this work, we investigated the rotation angles of 5° and 10° with respect to the EEG signal quality and the experience of the participants. The larger rotation angles provide more opportunities to the CME to fit the shape of the head. However, when the rotation angle is large, the CME may approach to the scalp with a tilted angle in the setup and provide pains to the participants.

3.3. Rotation angle of the ball joint (in 3. Experimental procedures)
Line 178:

The participants were nine healthy men and one healthy woman in their twenties. EEG measurements were conducted during eye closing, and the signals were acquired at O1 (left occipital part) in the international 10–20 system. For all the participants, both of the rotation angles of 5° and 10° were applied with the recording time of EEG for 30 s, where 5° and then 10° was tested for the first half of the participants and the order was reversed for the other half. In order to make the test conditions identical except for the rotation angle, we separated the hair manually under visual observation. The pressing force of the electrode was set to be identical for 5° and 10°. After the measurements with the two rotation angles, the participants were requested to answer which angle was less painful or bothering.

4.3. Comparison of the rotation angle of the ball joint (in 4. Results)
Line 220:

We adopted the signal-to-noise ratio (SNR) as signal quality and SNR was calculated as following equation (3) in this study.

Where PSD is the power spectral density of the raw EEG data. In this experiment, the original EEG data was divided into epochs with 2048 data points with 50 % overlap. The PSD estimate was applied each epoch and all epoch were averaged. In Eq. (3),  is the PSD for the alpha band, and  is the PSD for all frequency bands other than the alpha band. In this experiment, the PSD was obtained by Labchart 7 (ADInstruments, Australia, Sydney).

Table 2 shows the results. There was no significant difference in the average of SNR between the rotation angles of 5° and 10°. The measurement order did not make significant differences, either. No participants reported severe pain. 6 participants answered that the rotation angle of 10° provided more pain.

5.3. The rotation angle of the ball joint (in 5. Discussion)
Line 297:

As we described in Section 4.3, The rotation angles of 5° and 10° did not result in the change of the signal quality. No subjects answered that 5° was more painful than 10°. Subject 2 reported slight movement of the electrode at the rotation angle of 10°. We suspected that this was because the degree of freedom of the ball joint increased more than necessary. Subject 3 said that he felt more thorny feeling at the rotation angle of 10° than 5°. The thorny feeling is caused when the corner of the electrode holder contacted to the scalp. From above discussions, we concluded that the rotation angle of 5° was sufficiently good for EEG-Hat.

      2. if the EEG-Hat is the same technology, then please explain how do the new            results justify the originality of the paper

 As we described above, the design of the EEG-Hat was refined and its benefits were experimentally verified. The setup time was drastically reduced. The rotation angle of the ball joint was decided according to the SNR and the experiences of the participants. In addition, we carefully investigated the signals from all the channels. EEG signals were successfully acquired from the hairy parietal and occipital parts (See section 4.4). It was found the EEG-Hat had more difficulties in measuring EEG from the right frontal of the head (see Figure 11). We believe these results and discussion are sufficiently significant to claim the originality.

We comment on these in the following revised sentence:

1. Introduction
Line 67:

We firstly presented the concept of the EEG-Hat [46]. In this paper, we refined the design to facilitate the setup and enhance the reliability in the measurement. Experiments and thorough analyses were conducted to further verified the effectiveness of the EEG-Hat.

4.4. EEG measurement with the EEG-Hat
Line 236:

Table 3 gives the SNRs obtained during eye opening and eye closing. The SNR was calculated to verify that the EEG-Hat recorded meaningful EEG signals. EEG data spanning 30 s from 5 s after the start of the measurement were analyzed for both eye opening and eye closing. A comparison of the signals during eye opening and eye closing obtained from each channel for each subject demonstrates that a higher SNR was obtained during eye closing than eye opening for all signals except that from channel O2 for subject 4.

Figure 11 shows the PSD for each channel for 5 min during eye closure. From this 5-min PSD, the stability of the EEG measurement with EEG-Hat was obtained. The PSD for each channel for each subject was normalized with respect to its integral over the plotted range and averaged over all subjects. Large peaks corresponding to the alpha band (8–13 Hz) were observed in channels F3, Cz, O2, and O1. In F4, although the peak corresponding to the alpha band was not clear, the power in the alpha band was higher than that for frequencies exceeding 13 Hz.

6. Conclusion
Line342:

The rotation angles of the ball joint to adjust the angles of the CME to the scalp were tested and the rotation angle of 5° was experimentally selected as the suitable parameter taking the signal-to-noise ration and user experiences into consideration. EEG signals were stably acquired in five channels without manual hair separation while the setup time was drastically reduced from our prior work.

     3. consider to adjust the title

We adjusted the title:

Title:

Design and Characterization of EEG-Hat for Reliable EEG Measurement

     4. cite the above mention paper 

We cited the paper at following section and add the reference of the paper as [46].

1. Introduction
Line 67:

We firstly presented the concept of the EEG-Hat [46]. In this paper, we refined the design to facilitate the setup and enhance the reliability in the measurement. Experiments and thorough analyses were conducted to further verified the effectiveness of the EEG-Hat.

2.1 Module design
Line 105:

The lead wire is connected to the electrode via the inside of the insertion part, which prevented the lead wire from hindering movement of the shutter and contact of the electrode to the scalp. It was not the case for our prior work [46], where the design resulted in low measurement yield and long setup time.

References

[46] T. Kawana, Y. Yoshida, Y. Kudo, and N. Miki, “EEG-Hat with Candle-like Microneedle Electrode,” Proceedings of the Annual International Conference of the IEEE Engineering in Medicine and Biology Society, EMBS. IEEE, pp. 1111–1114, 2019.

In addition, since no biocompatibility study is presented and since the eventual application of these electrodes are for humans it could be interesting for the readers if the Authors can provide information of biocompatibility the materials used. 

The surface of the electrodes is covered with porous parylene that has high biocompatibility. Silver is used as the conductive layer, which is widely used for bioapplications.

We add this explanation in Introduction:

1. Introduction
Line 44:

The CME is covered with parylene which has sufficient mechanical strength and biocompatibility [22].

References

[22] S. Takeuchi, D. Ziegler, Y. Yoshida, K. Mabuchi, and T. Suzuki, “Parylene flexible neural probes integrated with microfluidic channels,” Lab Chip, vol. 5, no. 5, pp. 519–523, Apr. 2005.

Moreover, we would like to discuss the comfort for the participant in this study. Only in experiment 3.1 (Quantification of contact force and hair separation), 2 participants report discomfort for the pressing force. However, in the experiment of EEG measurement, no participant report sever pain.

In the experiment 3.1, participant 1 (15 N) and 4 (10 N) reported discomfort for the pressing force as the experiment progressed. However, the above forces were recorded when the hair was manually separated as much as possible and the impedance became below 30 kΩ. The impedance of approximately 100 kΩ is sufficient for EEG measurement with dry-electrode including CME [49]. Therefore, for the EEG measurement with the EEG-Hat, it is not necessary to press the electrodes until the participants feel discomfort when the hair was properly separated by the shutter. For EEG-Hat, the subjects can select a force for which they do not feel discomfort within the range of 0 to 16.3 N by changing the shrinkage of the spring in the insertion part.

In EEG measurement experiments (e.g. the experiments to determine the ball joint angle and EEG measurement from 5 pointes for 5.5 min), no severe pain was reported by the participants. The pain depends on the EEG-Hat design, which we newly reported in Table 2. In Table 2, no participants reported severe pain. We asked participant which of the rotation angle was, if anything, more painful. 6 participants answered that the rotation angle of 10° is more painful. We decided the rotation angle to be 5°.

The edges of the shutters were covered with plastic tapes, as shown Figure 1 (c), and does not hurt the scalp. Visual observation did not show any damages on the scalp.

We add these comments on these in following sentence:

2.2. Fabrication
Line 124:

The edges of the shutters are covered with plastic tapes, as shown Figure 1 (c), and does not hurt the scalp.

4.3. The rotation angle of the ball joint
Line 228:

Table 2 shows the results. There was no significant difference in the average of SNR between the rotation angles of 5° and 10°. The measurement order did not make significant differences, either. No participants reported severe pain. 6 participants answered that the rotation angle of 10° provided more pain. 

4.4. EEG measurement with the EEG-Hat
Line 261:

Visual observation did not show any damages on the scalp. No participants reported sever pain during the EEG measurement for 5.5 min.

5.1. Quantification of contact force and hair separation
Line 276:

Additionally, it was found that a pressing force in the range of 8–15 N is sufficient for EEG measurements with CMEs. From the result of pressing force, the spring in the insertion part was designed to have a spring constant of 1.36 N/mm. The designed modules can compress the spring in increments of 1.5 to 12 mm. With this design, the EEG-Hat could provide a contact force ranging from 0 to 16.3 N. In this experiment, participant 1 and 4 reported discomfort as the experiment progressed. The forces in the above range was recorded when the hair was manually separated as much as possible and the impedance was below 30 kΩ. The impedance of approximately 100 kΩ is sufficient for EEG measurement with dry-electrode including CME [49]. Therefore, for the EEG measurement with the EEG-Hat, it is not necessary to press the electrodes until the participants feel discomfort when the hair was properly separated by the shutter. For EEG-Hat, the subjects can select a force for which they do not feel discomfort within the range of 0 to 16.3 N by changing the shrinkage of the spring in the insertion part.

Reviewer 2 Report

This manuscript propose an EEG device with candle-like microneedle electrodes (CMEs), called the EEG-Hat that allows EEG signal recording without skin treatment or conductive gels. The EEG-Hat is novel, simple and interesting, and the results validated the feasibility. Therefore, I suggest for publication after addressing the following issues:

  1. The biosafety should be described. E.g., does the EEG-Hat cause pain and/or infection?
  2. The stability of impedance related to time and movement should be characterized.

Author Response

This manuscript propose an EEG device with candle-like microneedle electrodes (CMEs), called the EEG-Hat that allows EEG signal recording without skin treatment or conductive gels. The EEG-Hat is novel, simple and interesting, and the results validated the feasibility. Therefore, I suggest for publication after addressing the following issues:

  1. The biosafety should be described. E.g., does the EEG-Hat cause pain and/or infection?

We appreciate your precious comment. CME is coated with biocompatible parylene polymer. This description is added to 1. introduction.

1. Introduction
Line 44:

The CME is covered with parylene which has sufficient mechanical strength and biocompatibility[22].

Only in the experiment 3.1, participant 1 (15 N) and 4 (10 N) reported discomfort for the pressing force as the experiment progressed. However, the above forces were recorded when the hair was manually separated as much as possible and the impedance became below 30 kΩ. The impedance of approximately 100 kΩ is sufficient for EEG measurement with dry-electrode including CME [49]. Therefore, for the EEG measurement with the EEG-Hat, it is not necessary to press the electrodes until the participants feel discomfort when the hair was properly separated by the shutter. For EEG-Hat, the subjects can select a force for which they do not feel discomfort within the range of 0 to 16.3 N by changing the shrinkage of the spring in the insertion part.

In EEG measurement experiments (e.g. the experiments to determine the ball joint angle and EEG measurement from 5 pointes), participants did not report severe pain during the experiments of EEG measurement for 5.5 min. The pain depends on the EEG-Hat design, which we newly reported in Table 2. In Table 2, no participants reported severe pain. We asked participant which of the rotation angle was, if anything, more painful. 6 participants answered that the rotation angle of 10° is more painful. We decided the rotation angle to be 5°.

The edges of the shutters were covered with plastic tapes, as shown Figure 1 (c), and does not hurt the scalp. Visual observation did not show any damages on the scalp.

We add these comments on these in following sentence:

2.2. Fabrication

Line 124:

The edges of the shutters are covered with plastic tapes, as shown Figure 1 (c), and does not hurt the scalp.

4.3. The rotation angle of the ball joint
Line 228:

Table 2 shows the results. There was no significant difference in the average of SNR between the rotation angles of 5° and 10°. The measurement order did not make significant differences, either. No participants reported severe pain. 6 participants answered that the rotation angle of 10° provided more pain.

4.4. EEG measurement with the EEG-Hat
Line 261:

Visual observation did not show any damages on the scalp. No participants reported sever pain during the EEG measurement for 5.5 min.

5.1. Quantification of contact force and hair separation

Line 276:

Additionally, it was found that a pressing force in the range of 8–15 N is sufficient for EEG measurements with CMEs. From the result of pressing force, the spring in the insertion part was designed to have a spring constant of 1.36 N/mm. The designed modules can compress the spring in increments of 1.5 to 12 mm. With this design, the EEG-Hat could provide a contact force ranging from 0 to 16.3 N. In this experiment, participant 1 and 4 reported discomfort as the experiment progressed. The forces in the above range was recorded when the hair was manually separated as much as possible and the impedance was below 30 kΩ. The impedance of approximately 100 kΩ is sufficient for EEG measurement with dry-electrode including CME [49]. Therefore, for the EEG measurement with the EEG-Hat, it is not necessary to press the electrodes until the participants feel discomfort when the hair was properly separated by the shutter. For EEG-Hat, the subjects can select a force for which they do not feel discomfort within the range of 0 to 16.3 N by changing the shrinkage of the spring in the insertion part.

As to the concern about infection, the needle height is 200 um, which does not reach the pain point or epidermis, which would not cause any infection.

We add this comment in INTRODUCTION:

1. Introduction
Line 40:

The needle height is 200 um, which does not reach the pain point or epidermis, which would not cause any infection.

      2. The stability of impedance related to time and movement should be characterized.

We recorded the impedance at the beginning and ending of the experiments. We showed the results in Figure 12. Due to the limitation of the measurement system, we could not acquire both EEG and the impedance simultaneously during the experiments. According to Figure 12, Cz was the only channel at which the impedance increased 50 kΩ or more for all the participants. Each channel receives a reaction force against the pressing force of the electrode to the scalp, which results in deformation of the hat. The deformation at the position of Cz is considered to be the largest among all the measurement position. In other positions, during the EEG measurement for 5 min, the increase of the impedance was acceptable.

We did not characterize the effect of the movement of the participants. The heads of the participants were not fixed, however, they sit on a chair and did not make any actions. In our future work, we will look into the robustness of the EEG-Hat against the movement of the participants.

3.4. EEG measurement with the EEG-Hat (in 3. Experimental procedures)
Line 196:

The scalp-electrode impedance was measured at the start and the end of the measurement with the impedance meter (SIGGI II, Easycap, BRAIN VISION UK Ltd., London, UK).

4.4. EEG measurement with the EEG-Hat (in 4. Results)
Line 248:

Figure 12 shows the scalp-electrode impedance for each channel at the beginning and the end of the experiments. The measurement limit of the impedance meter was 200 kΩ. Since we could acquire sufficiently good EEG when the impedance was below 200 kΩ, in the experiments we set the acceptable impedance to be 200 kΩ. The × mark means that the impedance was over 200 kΩ. The increase of impedance was less 50 kΩ for 13 out of 20 channels and it was over 50 kΩ for 7 out of 20 channels. For Cz, an increase of over 50 kΩ was observed for all the participants. The impedance at O2 for all the participants remained below 100 kΩ.

5.4. EEG measurement with the EEG-Hat (in 5. Discussion)
Line 329:

According to Figure 12, Cz was the only channel at which the impedance increased 50 kΩ or more for all the participants. Each channel receives a reaction force against the pressing force of the electrode to the scalp, which results in deformation of the hat. The deformation at the position of Cz is considered to be the largest among all the measurement position. In other positions, during the EEG measurement for 5 min, the increase of the impedance was acceptable.

Reviewer 3 Report

This work by Kawana et al. reports the use of  microneedles integrated into a hat to be used as electroencephalogram measurement. Moreover, they present a shutter clamp module to separate hairs from permit better contact with the scalp. Although the use of microneedles to measure electrical potential fluctuations in the brain is well studied topic, the authors present a clever integration of different technologies that could help expand their use, based on the integration of the shutter clamps, needles and mechanical modules to adjust and maintain pressure. The article is well structured and presents attractive schematics that make it easy to understand the main claims of this works.

I Support publication on this work on Micromachines after these minor comments are addressed.

  1. The article would benefit by combining some figures together. For example, combining figure 2 and figure 6 would show both schematic and actual hat, making it more clear from the beginning on the actual topic of the research. Similarly, some schematic figures could be combined also, for example figure 4 and 5 both show the clamp mechanism for separating hair.
  2. The authors need to give a critical outlook in the conclusion section, on the challenges of this particular work and in general for the field of EEG monitoring

Author Response

  1. The article would benefit by combining some figures together. For example, combining figure 2 and figure 6 would show both schematic and actual hat, making it more clear from the beginning on the actual topic of the research. Similarly, some schematic figures could be combined also, for example figure 4 and 5 both show the clamp mechanism for separating hair.

We appreciate your precious comment. We combined the figures and changed the description.

      2. The authors need to give a critical outlook in the conclusion section, on the challenges of this particular work and in general for the field of EEG monitoring

We revised the conclusion as you suggested:

6. Conclusion
Line 346:
The fixation mechanism of EEG electrodes, such as the EEG-Hat, will play a critical role for the advancement of EEG research. Characterization of such mechanisms with respect to the SNR, the contact impedance, the setup time, and the user experience can be a good design guideline. This proposed guideline will contribute the development of the comfortable and reliable EEG-devices.

Reviewer 4 Report

Summary: The authors demonstrate design and application of an EEG hat for fixation of custom made candle-like microneedle electrodes (CMEs). A shutter mechanism to reduce hair occupancy from the placement site is also incorporate into the hat design which the authors have shown is critical to obtaining good signal to noise ratio (SNR). The shutter design (shutter angle) was optimized to ensure hair occupancy rate of less than 80%. The force required to fix the CMEs was determined which was incorporated in the EEG hat design by using a spring mechanism. 

Overall Comment: The EEG hat design which incorporates both the shutter mechanism to displace the hair and also a spring mechanism to fix the electrodes would help advance clinical studies, especially considering the fact that in the future, EEG could be used daily for applications such as Brain Computer Interfaces (BCIs). A hat design such as the one developed by the authors could improve patient compliance as it could be deemed more aesthetically pleasing and at the same time provide sufficient mechanism to obtain good SNRs. Given that the patient comfort/experience is a major factor for advancing such headsets, the authors could have obtained and presented the feedback on comfort from the human subjects (see Specific Comment 4 for more details)

Specific Comment 1: Line 37-38. The authors should rephrase the sentence that describes the dimensions of CMEs. A zoomed in image of a single needle in Figure 1 with dimensions would provide clarity to readers. Tip sharpness (or Tip radii) should also be included as it could potentially affect the forces measured.

Specific Comment 2: Table 1. The authors should add appropriate citations to each row in the Table 1.

Specific Comment 3: Line 143. The authors should add an explanation as to why only a 10 degree range (50-60 degrees) was chosen to determine the appropriate shutter angle.

Specific Comment 4: Line 204. Although the authors do provide the forces measured to ensure good SNR ratio, however it is not clear if this amount of force experienced by the user is comfortable. If the authors did collect such human factors data from the subjects, they should be presented. If such data is not available, force data from literature could be used to discuss if these force levels are acceptable or not.

Author Response

Overall Comment: The EEG hat design which incorporates both the shutter mechanism to displace the hair and also a spring mechanism to fix the electrodes would help advance clinical studies, especially considering the fact that in the future, EEG could be used daily for applications such as Brain Computer Interfaces (BCIs). A hat design such as the one developed by the authors could improve patient compliance as it could be deemed more aesthetically pleasing and at the same time provide sufficient mechanism to obtain good SNRs. Given that the patient comfort/experience is a major factor for advancing such headsets, the authors could have obtained and presented the feedback on comfort from the human subjects (see Specific Comment 4 for more details)

Specific Comment 1: Line 37-38. The authors should rephrase the sentence that describes the dimensions of CMEs. A zoomed in image of a single needle in Figure 1 with dimensions would provide clarity to readers. Tip sharpness (or Tip radii) should also be included as it could potentially affect the forces measured.

 We appreciate your precious comments. We revised the sentence as follows and figure as shown Figure 1.  

1. Introduction
Line 38:
The CME is composed of an array of protruding formations, which is a relatively thick pillar of 400 μm in diameter and 1000 μm in length. Each of pillar has a sharp tip of 200 μm in length and 200 μm in radius.

Specific Comment 2: Table 1. The authors should add appropriate citations to each row in the Table 1.

We add the citations in the table 1 (See Table 1) and more references:

References

[25]M. Clemente, A. Rodríguez, B. Rey, and M. Alcañiz, “Assessment of the influence of navigation control and screen size on the sense of presence in virtual reality using EEG,” Expert Syst. Appl., vol. 41, no. 4 PART 2, pp. 1584–1592, 2014.

[28] V. Zotev, R. Phillips, H. Yuan, M. Misaki, and J. Bodurka, “Self-regulation of human brain activity using simultaneous real-time fMRI and EEG neurofeedback,” Neuroimage, vol. 85, pp. 985–995, 2014.

[36]G. S. Taylor and C. Schmidt, “Empirical evaluation of the Emotiv EPOC BCI headset for the detection of mental actions,” in Proceedings of the Human Factors and Ergonomics Society, 2012, pp. 193–197.

[37] D. J. Lancheros-Cuesta, J. L. R. Arias, Y. Y. Forero, and A. C. Duran, “Evaluation of e-learning activities with NeuroSky MindWave EEG [Evaluación de actividades e-learning con NeuroSky MindWave EEG],” in Iberian Conference on Information Systems and Technologies, CISTI, 2018, vol. 2018-June, pp. 1–6.

[41]Y. Liu et al., “Implementation of SSVEP based BCI with Emotiv EPOC,” in Proceedings of IEEE International Conference on Virtual Environments, Human-Computer Interfaces, and Measurement Systems,VECIMS, 2012, pp. 34–37.

Specific Comment 3: Line 143. The authors should add an explanation as to why only a 10 degree range (50-60 degrees) was chosen to determine the appropriate shutter angle.

We tested three shutter angles by considering the size of the module and the ease of opening of the shutter. We added the discussion how we selected the value with a new figure as follows:

3.2. Determination of the shutter angle
Line 161:
These three shutter angles were selected by considering the size of the module and the ease of opening the shutter. The long shutter was unsuitable for a compact appearance of the device. Figure 8 (a) shows the relationship among the parameters of the shutter. The shutter height y in Figure 8 (a) is derived from the following equation (2) with shutter angle θ

 y = 12tanθ + 1.5/cosθ  (2)

According to equation (2), when shutter angle θ is over 65°, the shutter height y becomes over 30 mm and starts to increase rapidly. Therefore, we decided that the shutter angle θ was below 65°. Moreover, the shutter opens by the force applied by the contact with the insertion part as shown Figure 8 (b). In Figure 8 (b), when shutter angle θ is below 45°, the horizontal component of f become lager than the vertical component and opening of the shutter would become difficult. Therefore, the shutter angles θ of over 45° were selected. We tested the shutter angles θ of 50°, 55°, and 60° to find the most suitable one.

Specific Comment 4: Line 204. Although the authors do provide the forces measured to ensure good SNR ratio, however it is not clear if this amount of force experienced by the user is comfortable. If the authors did collect such human factors data from the subjects, they should be presented. If such data is not available, force data from literature could be used to discuss if these force levels are acceptable or not

Thank you for your precious comment. In this experiment, participant 1 (15 N) and 4 (10 N) reported discomfort for the pressing force as the experiment progressed. However, the forces in the above range was recorded when the hair was manually separated as much as possible and the impedance became below 30 kΩ. The impedance of approximately 100 kΩ is sufficient for EEG measurement with dry-electrode including CME [49]. Therefore, for the EEG measurement with the EEG-Hat, it is not necessary to press the electrodes until the participants feel discomfort when the hair was properly separated by the shutter. For EEG-Hat, the subjects can select a force for which they do not feel discomfort within the range of 0 to 16.3 N by changing the shrinkage of the spring in the insertion part. In EEG measurement experiments (e.g. the experiments to determine the ball joint angle and EEG measurement from 5 pointes), no severe pain was reported by the participants. Some references are also added:

5.1. Quantification of contact force and hair separation
Line 276:
Additionally, it was found that a pressing force in the range of 8–15 N is sufficient for EEG measurements with CMEs. From the result of pressing force, the spring in the insertion part was designed to have a spring constant of 1.36 N/mm. The designed modules can compress the spring in increments of 1.5 to 12 mm. With this design, the EEG-Hat could provide a contact force ranging from 0 to 16.3 N. In this experiment, participant 1 and 4 reported discomfort as the experiment progressed. The forces in the above range was recorded when the hair was manually separated as much as possible and the impedance was below 30 kΩ. The impedance of approximately 100 kΩ is sufficient for EEG measurement with dry-electrode including CME [49]. Therefore, for the EEG measurement with the EEG-Hat, it is not necessary to press the electrodes until the participants feel discomfort when the hair was properly separated by the shutter. For EEG-Hat, the subjects can select a force for which they do not feel discomfort within the range of 0 to 16.3 N by changing the shrinkage of the spring in the insertion part.

We did not collect the data on human factors, such as skin thickness, stiffness, etc. We added the references [47-49].

3.1 Quantification of contact force and hair separation
Line 143:
In this experiment, the scalp–electrode impedance was used as the evaluation index. However, this parameter is highly subject-dependent, in part because of the variability in human dermal characteristics [47]-[49]. For example, harder skin requires more force to lower the impedance.

References

[47] S. P. Davis, B. J. Landis, Z. H. Adams, M. G. Allen, and M. R. Prausnitz, “Insertion of microneedles into skin: Measurement and prediction of insertion force and needle fracture force,” J. Biomech., vol. 37, no. 8, pp. 1155–1163, 2004.

[48] H. Alexander and D.L.Miller, “Determing Skin Thickness with Pulsed Ultra Sound,” J. Invest. Dermatol., vol. 72, no. 1, pp. 17–19, 1979.

[49] K. Moronkeji, S. Todd, I. Dawidowska, S. D. Barrett, and R. Akhtar, “The role of subcutaneous tissue stiffness on microneedle performance in a representative in vitro model of skin,” Journal of Controlled Release, vol. 265. pp. 102–112, 2017.
